# The Orphan GPR50 Receptor Regulates the Aggressiveness of Breast Cancer Stem-like Cells via Targeting the NF-kB Signaling Pathway

**DOI:** 10.3390/ijms24032804

**Published:** 2023-02-01

**Authors:** Polash Kumar Biswas, Sang Rok Park, Jongyub An, Kyung Min Lim, Ahmed Abdal Dayem, Kwonwoo Song, Hye Yeon Choi, Yujin Choi, Kyoung Sik Park, Hyun Jin Shin, Aram Kim, Minchan Gil, Subbroto Kumar Saha, Ssang-Goo Cho

**Affiliations:** 1Department of Stem Cell and Regenerative Biotechnology, Molecular & Cellular Reprogramming Center (MCRC), Incurable Disease Animal Model & Stem Cell Institute (IDASI), Konkuk University, 120 Neungdong-ro, Gwangjin-gu, Seoul 05029, Republic of Korea; 2Division of Biological Sciences, University of Montana, Missoula, MT 59812, USA; 3Department of Surgery, Konkuk University Medical Center, Konkuk University School of Medicine, Seoul 05029, Republic of Korea; 4Department of Ophthalmology, Research Institute of Medical Science, Konkuk University Medical Center, Konkuk University School of Medicine, Seoul 05029, Republic of Korea; 5Department of Urology, Konkuk University Medical Center, Konkuk University School of Medicine, Seoul 05029, Republic of Korea

**Keywords:** GPR50, breast cancer stem cell (BCSC), NF-kB, Notch, ADAM17

## Abstract

The expression of GPR50 in CSLC and several breast cancer cell lines was assessed by RT-PCR and online platform (UALCAN, GEPIA, and R2 gene analysis). The role of GPR50 in driving CSLC, sphere formation, cell proliferation, and migration was performed using shGPR50 gene knockdown, and the role of GPR50-regulated signaling pathways was examined by Western blotting and Luciferase Assay. Herein, we confirmed that the expression of G protein-coupled receptor 50 (GPR50) in cancer stem-like cells (CSLC) is higher than that in other cancer cells. We examined that the knockdown of GPR50 in CSLC led to decreased cancer properties, such as sphere formation, cell proliferation, migration, and stemness. GPR50 silencing downregulates NF-kB signaling, which is involved in sphere formation and aggressiveness of CSLC. In addition, we demonstrated that GPR50 also regulates ADAM-17 activity by activating NOTCH signaling pathways through the AKT/SP1 axis in CSLC. Overall, we demonstrated a novel GPR50-mediated regulation of the NF-κB-Notch signaling pathway, which can provide insights into CSLC progression and prognosis, and NF-κB-NOTCH-based CSLC treatment strategies.

## 1. Introduction

Despite advances in biomedical research, breast cancer (BC) remains a major health concern. There were approximately 1.7 million new cases of BC worldwide in 2013, suggesting slow progress in the prevention setting [1]. In the last decade, BC has been a leading cause of increased mortality in women [2]. Despite advances in early detection methods and treatments such as chemotherapy, radiotherapy, and endocrine therapy, some BC patients experience poor drug response and tumor recurrence, and many do not survive. Therefore, improving patient outcomes requires a better understanding of the underlying mechanisms of BC [3].

Clarke et al. showed that breast cancer stem cells (BCSCs) play a major role in drug resistance and relapse of solid tumors [4]. BCSCs are a small population of BC cells that promote tumor progression and drug resistance to conventional therapy [5,6]. This study explored some aspects of BC related to G protein-coupled receptors (GPCRs), which are responsible for a variety of physiological and pathophysiological forms, including malignancy development. GPCRs also play a major role in intracellular and extracellular signaling and physiological functions of the body. Many members of this family of receptors remain esoteric, particularly orphan GPCRs that do not bind to a unique ligand. GPCRs mediate 80% of transmembrane signal transduction through external environmental factors such as light, heat, and temperature [7]. Therefore, GPCRs are considered important targets for the development of new drug treatments [8].

The G protein-coupled receptor 50 (GPR50) gene is a member of the GPCRs implicated in antiproliferative effects, and poor survival prognosis is related to the low expression of GPR50 in breast cancer. This correlation highlights GPR50 as a diagnostic tool and its potential use as a therapeutic target for cancer [9].

It has been reported that GPR50 can make a heterodimer with type I TGFβ receptor (TβRI) to lower the tumor growth in the human breast by activating the antiproliferative effect of the TGFβ receptor. Basically, TGFβ works through two kinases, TβRI and TβRII, by attaching TGFβ to TβRII to recruit TβRI. This results in the detachment of the FKBP12 inhibitor and the joining of Smad2/3 proteins [10,11,12]. This ectopic expression of GPR50 protects against tumor development, and its absence is evident in pro-tumorigenic animal models. Therefore, it can be predicted that GPR50 might help in better diagnosis and understanding in the case of BC [13]. A previous study reported that several GPCRs, including GPR50, are involved in the reprogramming of somatic cells to BCSCs, suggesting that GPR50 can act as a tumor suppressor in hepatocellular carcinoma (HCC) through the ADAM17-Notch signaling pathway [14,15].

However, little is known regarding the role of GPR50 in the regulation of breast cancer stem-like cells. GPR50 protein might be involved in the regulation of breast cancer, which might be involved in sphere formation, migration, and stemness of breast cancer stem-like cells. In this study, we investigated whether GPR50 knockdown decreased proliferation, migration, and sphere formation in breast cancer stem-like cells. We also tested whether GPR50 regulates NF-kB signaling. In this study, we showed that GPR50 directly interacts with Notch signaling. NF-kB signaling is downstream of Notch signaling and regulates the expression of BCL-2 in cancer stem-like cells (CSLC). GPR50 also regulates ADAM-17 activity by activating NOTCH signaling pathways through the AKT/SP1 axis in CSLC.

## 2. Results

### 2.1. CSLC Conversion and Gene Expression Changes in Tissues of Breast Cancer Patients Who Received Chemotherapy

The gene expression of CSLC with CD133high/CXCR4high/ALDH1high characteristics from breast cancer patients who received chemotherapy at Konkuk University Hospital was compared in various cancer cells, including breast cancer cells (Figure 1A) [16]. By comparing the genes (COL1A1, PDL2) involved in cancer cell progression, the expression level of GPR50 was found to be upregulated in CSLC (Figure 1B). Next, the UALCAN dataset confirmed the high expression of GPR50 in the tissues of triple-negative breast cancer (TNBC) patients (Figure 1C). We also compared the differences in GPR50 transcript (TPM) levels and survival rates between the general population and patients with BC. Our GEPIA data showed that the expression level of GPR50 was higher in patients with BC than in their normal counterparts (Figure 1D). In addition, the survival rate graph of breast cancer patients with high GPR50 levels on the R2 gene analysis platform showed a lower survival rate (Figure 1E). Additionally, the expression level of GPR50 regulates clinical prognosis of breast cancer. Additionally, it was discovered that breast cancer patients with poor DMFS and OS had increased GPR50 expression (Figure 1F,G). This result suggests that the prognosis is poor when the expression of the GPR50 gene is higher in BC patients than in the general population.

### 2.2. Change in Cancer Cell Aggressiveness According to the Expression Level of GPR50 in CSLC

Previous studies have shown that GPR50 expression regulates HCC [15]. As GPR50 expression in CSLC is significantly higher than that in other breast cancer cells, we first investigated the properties of CSLC that can be regulated by GPR50 expression. Mock (empty vector) and short hairpin (sh) GPR50 were delivered to CSLC to downregulate its expression to obtain EV (empty vector)-CSLC and shGPR50-CSLC. We confirmed that GPR50 was downregulated (Figure 2A). ShGPR50-mediated decreases in spheroid formation capacity (Figure 2B), spheroid size (Figure 2C), and stemness markers (Figure 3D,E) were observed. In addition, cell viability decreased by approximately 20% (Figure 2F), cell proliferation decreased by 70% at passage 4 (Figure 2G), and cell migration capacity was reduced by half at 48 h (Figure 2H,I) [17,18]. Cellular properties of cancer stem cells (proliferative capacity, viability, metastasis, spheroid formation, and stem cell marker gene expression) were altered. These results suggest an important role for GPR50 in CSLC.

### 2.3. GPR50 Regulates the Activity of NF-kB

Previous studies have shown that GPR50 regulates BC cells through Notch and AKT/SP1 axis signaling [15,19]. Based on this, the regulatory mechanism of NF-kB is the key to cancer cell survival, such as tumor survival, chemotherapy resistance, metastasis, EMT, and invasion, while maintaining a high level in tumor initiation cells (TICs) [20]. NF-kB is present in the cytoplasm in an inactive state in response to three inhibitors (IKbα, Ikbβ, and IKBγ) [21]. These three inhibitors do not function via phosphorylation. Our study identified the regulatory mechanism of NF-kB signaling in CSLCs. Using an online platform, we also examined chemicals related to the NF-kB and GPR50 signaling pathway (Appendix A).

Compared with EV-CSLC, phospho-IKBα/IKBα was decreased in shGPR50-CSLC, along with phospho-NF-kB/NF-kB (Figure 3A). Luciferase assay confirmed that the gene expression of NF-kB was decreased in each cell line (Figure 3B). Expression of the pro-survival BCL2 gene was confirmed to be reduced (Figure 3C) [22]. As a result, it was confirmed that the gene and protein of BCL 2, a survival-related factor, decreased, and that of BAX, an apoptosis-related factor, increased (Figure 3D). The effect of NF-kB activity on CSLC was confirmed by treatment with an NF-kB inhibitor (BAY 11-7081). The activity of NF-kB was reduced following treatment with BAY11-7081 (Figure 3E). Next, we confirmed whether NF-kB was involved in CSLC proliferation and spheroid formation. As a result, the experimental group treated with BAY showed reduced proliferation by about 20% compared to CSLC, which was not treated with anything, and was not significantly different from the CSLC-shGPR50 experimental group. However, significant differences were observed between shGPR50-CSLC and shGPR50-CSLC + BAY 11-7081 (Figure 3F). In addition, it was confirmed that the spheroid formation ability was reduced in CSLC. The size of the spheroids decreased by approximately 30% to 186 μM for EV-CSLC and 121 μM for shGPR50-CSLC. This suggests that GPR50 plays an important role in spheroid maintenance (Figure 3G,H) [23].

### 2.4. In CSLC, GPR50 Regulates the Expression of ADAM17 and, Consequently SP1/AKT Signaling

In a previous study, we reported that GPR50 plays an important role in regulating HCC progression through SP1/AKT and Notch signaling in HCC cells. Therefore, we identified the SP1/AKT signaling pathway. Among the SP families, SP1 and SP4 mRNA levels were reduced (Figure 4A). In addition, the p-AKT/AKT ratio decreased at the protein level (Figure 4B). Our results showed that the SP1/AKT signaling pathway decreased when GPR50 was knocked down [15]. Consequently, the transcriptional and translational signals of ADAM17 were downregulated. This supports GPR50 modulating of SP1/AKT signaling (Figure 4C) [24].

### 2.5. GPR50 Regulates the Expression of HES1 through the Notch Signaling Pathway in CSLCs

A decrease in ADAM17 expression was confirmed in a previous study. ADAM17 is a Notch regulator [25]. Notch is an important signal that affects cell proliferation, metastasis, and stemness. The effect of low ADAM17 expression, one of the factors that activates Notch, was also confirmed [25]. Among the various genes (HES1, MAML, RBP-J) regulated by Notch, the expression of HES1 was downregulated (Figure 5A) [26]. HES1 expression is involved in NF-kB activity through IKBα degradation [27]. Comparing the protein levels of HES1 and Notch1, the expression of Notch 1 was maintained, and only HES1 decreased (Figure 5B,C). The active form of Notch was decreased in shGPR50-CSLC. In support of this, when EV-CSLCs were treated with the Notch1 signal inhibitor (DAPT), the expression of HES1 was downregulated (Figure 5D). Subsequently, it was confirmed that NF-κB expression was reduced (Figure 5E). Cell viability and spheroid-forming ability were compared to the previous effect on cells, as NF-kB was downregulated. Both EV-CSLC and shGPR50-CSLC were significantly reduced when DAPT was used. However, EV-CSLC showed lower viability than shGPR50-CSLC when treated with DAPT (Figure 5F,G) [28].

## 3. Discussion

GPR50 is an orphan GPCR, as its ligand is unknown [29]. The previously known GPR50 is a cell membrane protein that is closely related to the melatonin receptor. GPR50, unlike general GPCR, has no known ligand, seven transmembrane domains in the cell membrane, and a long intercellular carboxyl-terminal domain (CTD) (~300 aa) from the inside of the cell [30]. Previous studies have shown that these CTD functions are involved in TβRI activation, glucocorticoid receptor signaling, Notch signaling, and Wnt/β-catenin signaling [9,31]. Studies have been conducted to prevent cancer development through GPR50 signaling of the TGF-β receptor. This indicates that GPR50 may act as a tumor suppressor. In addition, previous studies have revealed that GPR50 is a key gene that can treat cancer through Notch signaling in various hepatoma cells [15,30].

This study found that GPR50 expression was higher than that of other genes in CSLC-converted BC cell lines obtained from chemotherapy-treated patients, suggesting that GPR50 may be involved in breast cancer stem cell regulation. Therefore, we performed a study on NF-kB signaling related to the GPR50 gene in CSLC. The results of this study may be helpful in the treatment of cancer stem cells, a type of tumor-initiating cell.

By examining the differences in the characteristics of CSLC, we found that GPR50 is highly expressed in the CSLC cell line and confirmed that stemness markers and spheroid formation ability were reduced in the shGPR50 CSLC cell line. Previous studies have indicated that the aggressiveness and drug resistance of cancer stem cells and tumor-initiating cells (TIC) and their ability to form spheroids are involved in the expression and proliferation of stem genes [32,33]. We confirmed that the spheroid formation ability was reduced in the shGPR50-CSLC cell line. In addition, our findings showed a reduction in proliferation migration capacity after GPR50 knockdown, and showed that the stemness marker genes and proteins were downregulated (Figure 2). These findings suggest that GPR50 is significantly involved in CSLC properties.

Our results indicated that GPR50 regulated NF-kB in CSLC and elucidated its regulatory mechanism. NF-kB is widely known to play an essential role in cell survival, inflammation, and immunity. It plays a crucial role in the self-renewal ability, drug resistance, and recurrence of various cancer cells, including breast cancer stem cells [34,35,36]. Consequently, we identified several signals regulated by GPR50. When GPR50 was knocked down, the mechanism of low regulation of HES1 expression by Notch signaling was confirmed. HES1 is known as a protein that functions to stabilize IKBα [27]. Therefore, the decrease in p-IKBα among the results of this study can be explained as a phenomenon caused by the decrease in HES1 (Figure 3 and Figure 5). A decrease in HES1 leads to a reduction in p-IKBa and, consequently, a reduction in NF-kB activity. As a result, the cell survival-related protein BAX increased and BCL2 decreased. By increasing the BAX/BCL2 ratio, it was confirmed that GPR50 was involved in the survival of CSCL (Figure 3). These signaling pathways are summarized in Figure 6.

The relevance of GPR50 in cancer cell progression has been demonstrated in this and previous studies [29]. Furthermore, in our earlier study, cells from breast cancer patients were shown to be stimulated to convert into CSLCs by mimicking shear stress following blood flow in the body [16]. We confirmed that GPR50 in these CSLCs is highly regulated in the CSLCs in our study. The part of GPR50 that regulates the overall signaling of cancer stem cells is not yet clear. Consequently, inhibition of GRP50 suppresses cancer stem-like cell proliferation and cancer cell properties. In addition, NF-kB confirmed that GPR50 is a factor that regulates the above signal through AKT/Notch1/SP1 signaling. This suggests that GPR50 may be a relapse-free treatment option that targets and suppresses TIC in BC patients.

## 4. Materials and Methods

### 4.1. Cell Culture

Human cell lines for various cancer screening (MDA-MB231, CSLCS, J82, and T24) and embryonic kidney cells (HEK293T) were obtained from the American Type Culture Collection (ATTC, Manassas, VA, USA). All cell lines and BC patient-derived CD133high/CXCR4high/ALDH1high high sphere-forming CSLCs were cultured in high glucose Dulbecco’s modified Eagle medium or Roswell Park Memorial Institute medium (DMEM or RPM, respectively) 1640 (Sigma-Aldrich, Saint Louis, MO, USA) with 10% fetal bovine serum. The cells were then incubated in a humidified incubator with 5% CO2 at 37 °C. All the cell lines were subjected to short tandem repeat profiling for authentication. The BioMycoX Mycoplasma PCR detection kit (Cat. No. CS-D-25) (Cellsafe, Yeongtong-gu, Suwon, Republic of Korea) was used to test mycoplasma contamination in the cells. CD133high/CXCR4high/ALDH1high CSLCS were collected from breast tumor tissue samples. Data were collected from patients treated with chemotherapy who underwent mammotomy for breast tumors at the Breast Cancer Center, Konkuk University Hospital, Seoul. The study was conducted with approval from the institutional review board (IRB, KUH, 1020003).

### 4.2. Total RNA Extraction and RT-PCR Analysis

The Easy-Blue total RNA extraction kit (iNtRON Biotechnology, Gyeonggi-do, Republic of Korea) was used to collect total RNA from the cells, and the total RNA concentration was measured using a Nanodrop (ND1000) spectrophotometer (Nanodrop Technologies Inc., Wilmington DE, USA). Following that, 2 µg of total RNA was used to produce cDNA using M-MLV reverse transcriptase (Promega, Madison, WI, USA) according to the manufacturer’s instructions. The products were analyzed on 1% or 1.5% agarose gels after PCR. Quantitative real-time RT-PCR was performed using a PTC-200 thermal cycler reaction with a Chromo4 optical detector (MJ Research/Bio-Rad, Hercules, CA, USA) and Fast SYBR Green Master Mix (Applied Biosystems, Stockholm, Sweden). GAPDH, a housekeeping gene, was used to normalize the mRNA [37].

### 4.3. Western Blotting Procedure

Lysis buffer (1% Triton X-100 (Sigma-Aldrich), 100 mM Tris-HCl (pH 7.5), 10 mM NaCl, 10% glycerol (Amresco, Solon OH, USA), 50 mM sodium fluoride (Sigma-Aldrich), 1 mM phenylmethylsulfonyl fluoride (PMSF; Sigma-Aldrich), 1 mM p-nitrophenyl phosphate (Sigma-Aldrich), and 1 mM sodium orthovanadate (Sigma-Aldrich)) was used to lyse the cells, and the cell lysates were centrifuged at 13,000 rpm for 15 min at 4 °C. Bradford protein assay reagent (Bio-Rad) was used to quantify the amount of protein in the supernatant, and 10% or 12% sodium dodecyl sulfate-polyacrylamide gel electrophoresis resolved these proteins. The separated proteins were then placed on a nitrocellulose paper membrane (Bio-Rad), where they were blocked with 5% skimmed milk in Tris-buffered saline for 1 h. Following that, the appropriate primary antibodies were used to incubate the membranes against GPR50 (14032S, 1:1000), NANOG (SC33759, 1:300), OCT3/4 (SC-5279, 1:300), SOX-2 (SC-20088, 1:300), IKKa (19930S, 1:1000), IKBa (4814S, 1:1000), p-IKBa (2859S, 1:1000), p65 (D14E12, 1:1000), p-p65 (3033S, 1:1000), BAX (CST2772S, 1:1000), NOTCH1 (SC373891, 1:1000), T-AKT (92112, 1:800), p-AKT (SC-29315, 1:300), BCL-2 (SC-492, 1:300), TACE (SC-6416, 1:300), a-tubulin (SC-23975, 1:300), HES1 (SC-13844, 1:500), b-actin (SC-47778, 1:1000) (Santa Cruz Biotechnology, Dallas, TX, USA), and intracellular domain (ab83232, 1:1000) (Abcam, Cambridge, UK) overnight at 4 °C. A 2 h incubation was followed by rabbit (SC-2004), anti-mouse (SC-2005), -goat (SC-2020), or -rat (SC-2006) IgGs (1:1000) that were tagged with horseradish peroxidase (Santa Cruz Biotechnology). A high-performance chemiluminescence kit (Amersham Bioscience, Piscataway, NJ, USA) was used to track protein signals.

### 4.4. Cell Proliferation, Wound Healing/Migration Assays

Cell proliferation assay was performed as previously described. For the cell proliferation analysis, 1 × 10^4^ cells/well were seeded in 96-well plates. In the wound healing/migration assay [38], 1 × 10^6^ cells were seeded in 60 mm plates and grown until 90% confluence. Earlier, the cells were incubated with mitomycin C (10 µg/mL), which inhibited cell division. A 5% CO_2_ supply system and at 37 °C the cells were incubated, and after the incubation period, the monolayer of the cells was wounded with a 200 µL pipette tip. Cells were then washed thrice with phosphate-buffered saline (PBS). Nikon eclipse TE2000-U microscopy (Nikon Instruments Inc., Melville, NY, USA) captured the images of the wound at various time points (0, 12, and 24 h).

### 4.5. Sphere-Formation Assay

Cells (1 × 10^6^) were cultured and suspended in either growth medium (DMEM/RPM1 1640 (Sigma-Aldrich) with 10% fetal bovine serum) or SM (sphere-formation medium: serum-free DMEM/F12 supplemented with B27-supplement (1:50; Invitrogen, Waltham, MA, USA) supplemented with 20 ng/mL epidermal growth factor (Sigma-Aldrich), 10 μg/mL insulin (Invitrogen), and 0.4% bovine serum albumin (Sigma-Aldrich) using non-coated plates. To generate spheres in vitro, they were gathered by gentle centrifugation, dissociated into single cells [39], and then cultured to obtain next-generation spheres. Spheres obtained from CSLCs were collected after five days of culture. The culture was then stained with crystal violet (Sigma-Aldrich), and a Nikon eclipse TE2000-U microscope (Nikon Instruments Inc.) was used to capture the images before colony counting.

### 4.6. Luciferase Reporter Assay

CSLC cells (1 × 10^5^) were seeded in a 24-well plate to test the dual luciferase reporter assay and transiently transfected with NFkB: Renilla: shGPR50 vector (ratio 0.1:0.9:0.01) (a kind gift from Professor Keun Il Kim (Sookmyung Women’s University) using Lipofectamine 2000 reagents (1:3 ratio) (Invitrogen) [40]. After 24 h post-transfection history, luciferase activity was examined using a luciferase assay system (Promega) using a luminometer (Veritas microplate luminometer, Turner Biosystems, CA, USA), and β-galactosidase expression was used to normalize transfection efficiency.

### 4.7. Analysis of GPR50 Protein Expression in Various Types of TNBC

GPR50 protein expression levels in several TNBC types were investigated using the UALCAN database (Preston, Lancashire, UK) (http://ualcan.path.uab.edu, accessed on 28 October 2022). GPR50 transcription has been identified in various types of TNBC.

### 4.8. GEPIA Database Analysis

Online bioinformatics tool for the genome of cancer gene expression profiling Interactive analysis 2 (GEPIA2) (Peking University, Beijing, China) (http://gepia2.cancer-pku.cn accessed on 28 October 2022) was used to analyze RNA expression (TCGA) data. In this study, GEPIA2 was used to analyze the association between the expression level of GPR50 in TNBC and normal tissues.

### 4.9. Analysis of GPR50 Protein Expression in Various Types of TNBC

GPR50 protein expression in several TNBC types was investigated using the UALCAN database (Preston, Lancashire, UK) (https://ualcan.path.uab.edu/index.html accessed on 28 October 2022). Protein expression levels of GPR50 have been identified in various TNBC types.

### 4.10. R2: Genomic Analysis and Visualization Platform

The R2 platform (Academic Medical Center (AMC), Amsterdam, Netherlands) (https://hgserver1.amc.nl/cgi-bin/r2/main.cg accessed on 28 October 2022) is a publicly available website. In this study, we performed a survival analysis of the mRNA expression of the GPR50 gene in TNBC using the R2 online tool.

## Figures and Tables

**Figure 1 ijms-24-02804-f001:**
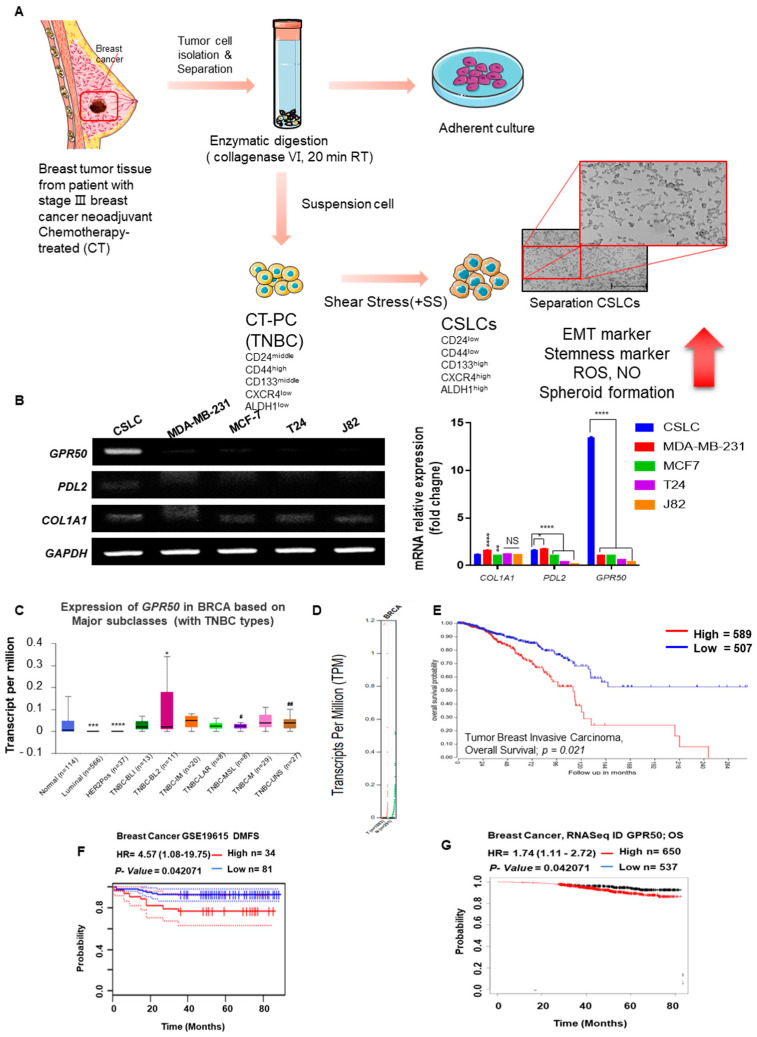
KU−CSLCs derived from patient suspected breast tumor (Chemotherapy treated). (**A**) Tissues were provided from humans who received chemotherapy. After separating the donated tissue, adhered cells and non-adherent cells were separated. By applying shear stress to the separated cells, conversion was performed by CSLC. This figure was made from a template licensed under the Creative Commons Attribution 3.0 Unported License: https://smart.servier.com. (**B**) Comparison of differences in gene expression levels associated with cancer progression in various cancer cell lines, including BC. (**C**) GRP50 expression in various cancer cells and CSLC (*p*-value * < 0.05, ** < 0.01, *** < 0.001, **** < 0.0001 vs. Normal; # < 0.05, ## < 0.01, vs. HER2Pos). (**D**) Differences in GPR50 transcript (TPM) between the general population and BC patients. (**E**) Differences in survival rate graphs according to GPR50 expression in BC patients (*p*-value * < 0.05, ** < 0.01, *** < 0.001, **** < 0.0001). The survival curve comparing patients with (**F**) high/high (red), low/low (blue), and (**G**) high/high (red), low/low (blue) in breast cancer. The clinical outcomes data were retrieved from (**F**) PrognoScan and (**G**) Kaplan−Meier plotter database (Information indicating a significant *p*-value is <0.05 and a non-significant *p*-value is expressed as ‘NS’ in graph).

**Figure 2 ijms-24-02804-f002:**
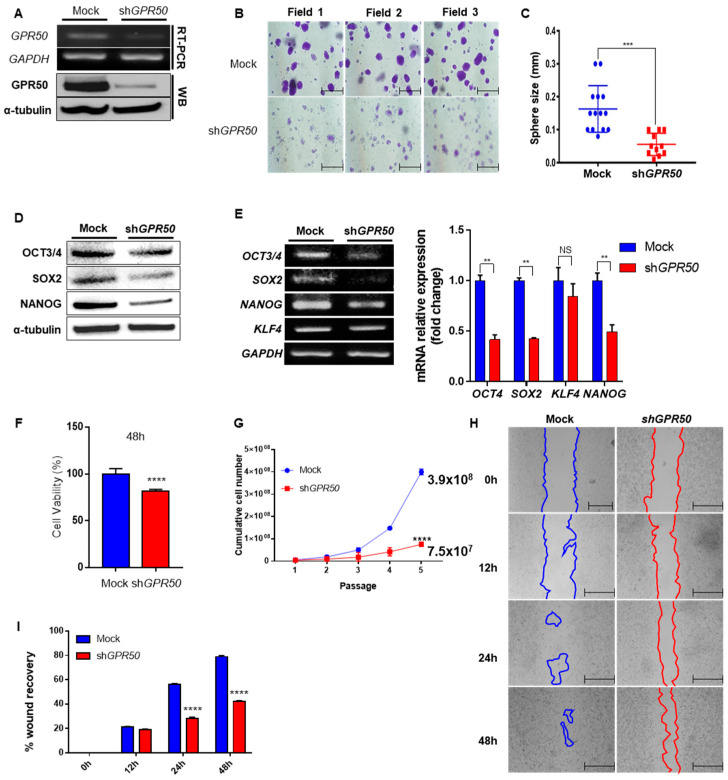
Knockdown of GPR50 represses cancer properties in breast cancer stem-like cells. (**A**) After transduction with Empty Vector (Mock) and shGPR50 in CSLC cells, GPR50 expression was confirmed by RT-PCR and Western blotting. (**B**,**C**) Spheroid formation pictures and size measurements of EV-CSLC and shGPR50-CSLC (Scale bar: 500 µm; n = 11~14). (**D**) Differences in stemness marker protein expression (OCT3/4, SOX2, NANOG) (**E**) The difference in stemness gene expression levels in EV-CSLC and shGPR50-CSLC (OCT3/4, SOX2, NANOG, KLF4). (**F**) Difference in cell viability through MTT assay (48 h). (**G**) Cumulative cell number (CPDL) graph for each passage through cell counting (4 passages). (**H**,**I**) Change in the migration ability of CSLCs through wound recovery experiments is shown in the photo. The degree of wound recovery is expressed in % and expressed as a graph (Scale bar: 200 µm) (*p*-value, ** < 0.01, *** < 0.001, **** < 0.0001, NS > 0.05).

**Figure 3 ijms-24-02804-f003:**
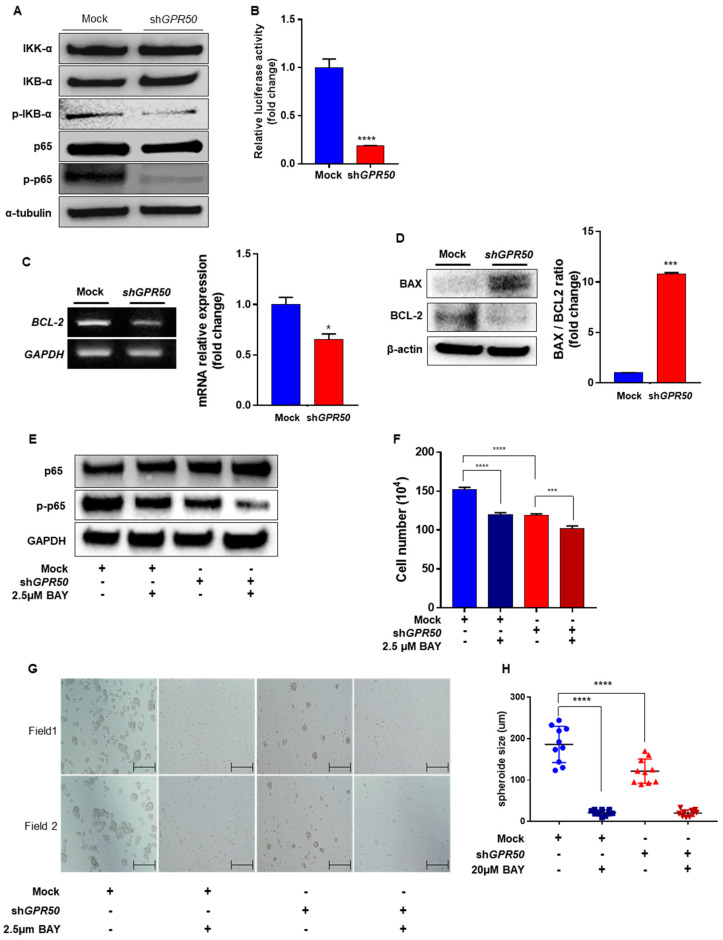
Silencing of GPR50 downregulates NF−kB signaling pathway in breast cancer stem−like cells. (**A**) Western blotting assay for NF−kB related proteins and NF−kB activity. (**B**) NF−kB expression was measured using a luciferase assay. (**C**) Confirmation of differences in BCL−2 gene expression level through qPCR. (**D**) Comparison of expression levels of cell survival−related proteins (BCL2, BAX). (**E**) Western blotting analysis of p−p65 expression level treats NF−kB inhibitor (BAY) and GPR50 knockdown. (**F**) Cell counting analysis was measured using trypan blue after treatment with NF−kB inhibitor (BAY). (**G**,**H**) comparison of spheroid formation ability changes and spheroid size measurement and graph (Scale bar: 500 µm; n = 10) (*p*-value * < 0.05, *** < 0.001, **** < 0.0001).

**Figure 4 ijms-24-02804-f004:**
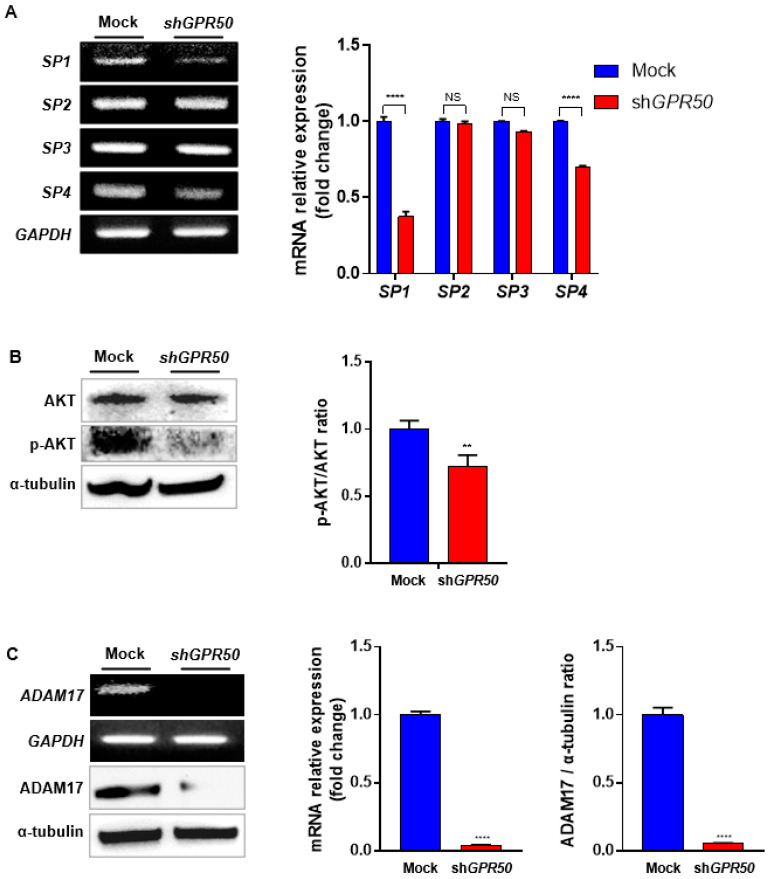
GPR50 controls the AKT/SP1 axis, which controls ADAM17 transcription. (**A**) Differences in expression levels of SP family (SP1, SP2, SP3, SP4) of EV-CSLC and shGPR50-CSLC. (**B**) AKT, p-AKT, and ERK expression levels were confirmed using Western blotting. α-Tubulin was used as a control. (**C**) Protein and mRNA levels of ADAM17 were compared by RT-PCR and Western blotting (*p*-value ** < 0.01, **** < 0.0001, NS > 0.05).

**Figure 5 ijms-24-02804-f005:**
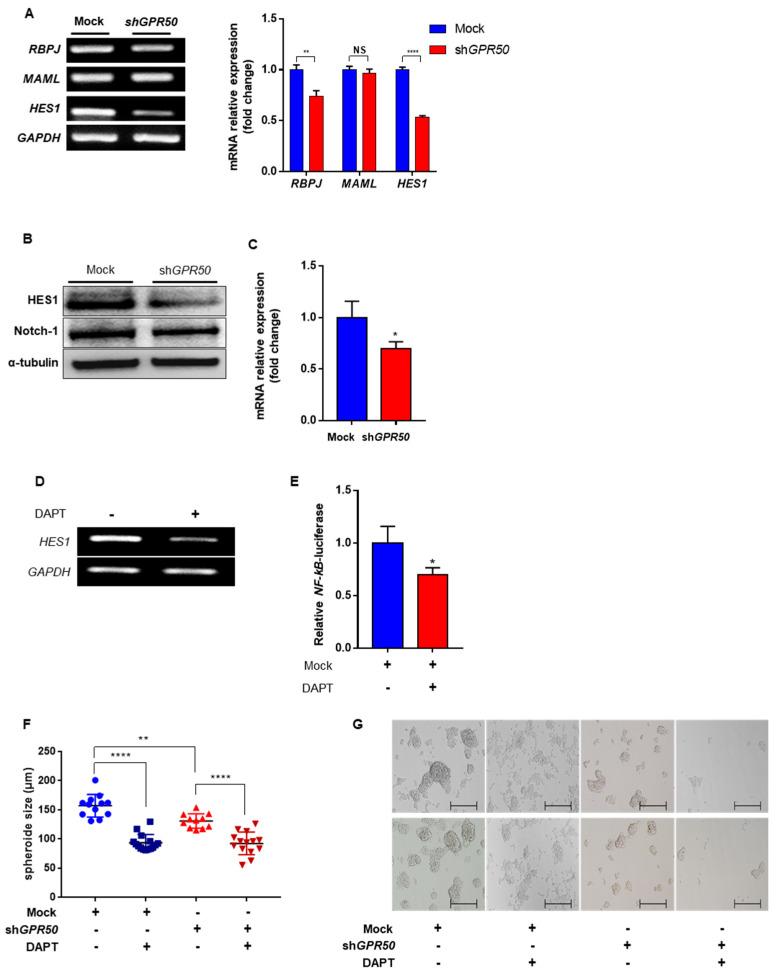
GPR50 modulates Notch Signaling Pathways in CSLC. (**A**) RNA expression level of HES1, MAML, and RBPJ genes. (**B**) Western blotting of HES1 and Notch1. (**C**) Comparison of mRNA expression of HES1. (**D**) HES1 expression level of PCR treated with Notch1 inhibitor (DAPT) in EV−CSLC. (**E**) Changes in NF−kB luciferase activity after DAPT treat in CSLC (DAPT concentration was used at 20 µM). (**F**) Comparison of cell viability according to DAPT treatment (n = 11~15). (**G**) Confirmation of spheroid formation ability according to DAPT treatment through a microscope (Scale bar: 500 µm) (*p*-value * < 0.05, ** < 0.01, **** < 0.0001, NS > 0.05).

**Figure 6 ijms-24-02804-f006:**
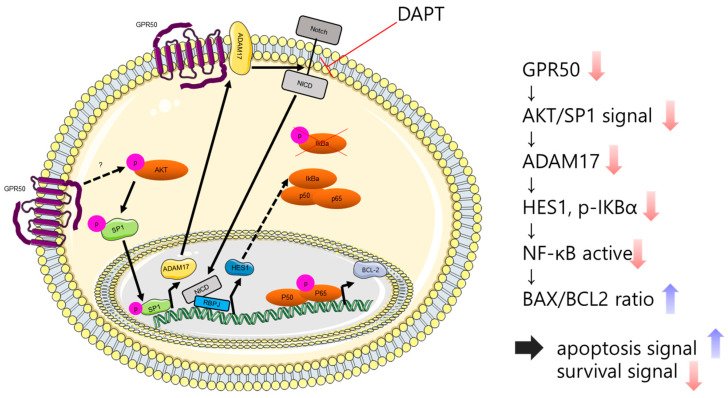
Schematic showing of NF-kB signaling regulated by GPR50 in CSLC. GPR50 regulates ADAM17 transcription through the AKT/SP1 axis, thereby regulating downstream HES1, and consequently, NF-kB activity. However, the mechanism of p-AKT has not yet been elucidated in detail. Depending on the activity of NF-kB, it is involved in CSLCs progression. This figure is made from a template licensed under the Creative Commons Attribution 3.0 Unported License: https://smart.servier.com.

## Data Availability

Not applicable.

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
