# Peer review of "The Orphan GPR50 Receptor Regulates the Aggressiveness of Breast Cancer Stem-like Cells via Targeting the NF-kB Signaling Pathway"

_ijms, 2023, doi:10.3390/ijms24032804_

Round 1
Reviewer 1 Report
In this manuscript the authors identified the expression level of GPR50 increased in the cultured CD133high /CXCR4high /ALDH1high cancer stem-like cells (CSLC) isolated from breast cancer patients who received the neoadjuvant chemotherapy. And the CSLC with knocked-down GPR50 presented lower cancer stem cell properties via NF-kB pathway. However, in the present form, it cannot be accepted for publication in IJMS because of the presence of several weak issues that need to be clarified/resolved.
1. The clinic information about the breast cancer patients who received chemotherapy are necessary to provide including the details on cancer invasion condition (such as typical H&E pathological staining figures, IHC markers, and so on) and the neoadjuvant chemotherapy process. How many tissue samples have been involved in the analysis? Can the increasing expression of GPR50 be induced by the neoadjuvant chemotherapy or not?
2. in Figure 1B, the GPR50 expression levels in the breast tumor tissues before inducing to CSLC are necessary to be detected to compare with that of the CSLC cells. The data in Figure 1B showed the differences of some gene expression levels in various cancer cell lines. It cannot conclude that GPR50 expression is associated with cancer progression or the expression level of GPR50 was found to be highly regulated. This conclusion is over-estimated.
3. The collation analysis between GPR50 expression and survival condition of BC, especially TNBC patients would be further analyzed by use of all the data from databases, such as ATCG.
4. In Figure 3, the transcriptomics data on the expression pattern of the CSLC and the GPR50-knocked-down CSLC are necessary to support the GPR50 regulating signal pathways, especially the expression changes of NF-kB pathway related molecules by systematic bioinformatic analysis.
5. In the spheroid formation experiments, the data on the number of colonies are necessary to be present in the manuscript.
6. The qualities of some Western blots are not good enough. It is recommended to provide the original photos of the SDS-PAGE gels or the PVDF films (including Markers) for all the WB results and add quantitative results.
7. Writing errors: Fig5: DAPT 20uM?
Author Response
Again, many thanks for the insightful and helpful criticism from the reviewers. We really hope that the editor and the reviewers will be pleased with the revised text and our replies to all of their insightful criticism.
We have responded with additional files.

Reviewer 2 Report
General comments
- In the area of study on breast cancer, the title "The orphan GPR50 receptor regulates the aggressiveness of breast cancer stem-like cells via targeting the NF-kB signaling pathway" is particularly fascinating because it offers more alternative genes used as a potential target for breast cancer prevention and treatment.
- The methodology, result, discussion, and introduction are all clearly stated.
Minor comments
- The statistical significance or P-value for Figures 1B and C, and Figure 2C should be included to demonstrate their significant upregulation.
- Include a brief explanation of how to calculate the assay's wound closure in the methodology section.
Author Response

(The authors gave the same response as above.)

Round 2
Reviewer 1 Report
1. Fig.3A: It’s hard to say the WB band of p-P65 is coincident with that (pp65) on the original data.
2. Figure titles for Fig.1 to Fig.5 are strongly suggested to add in to empathize the result.
3. Key words: Cancer stem cell (CSC) is suggested to Breast Cancer stem cell.
Author Response
Thanks again for your comment.
Responses to comments are attached below.
